# Lactobacillus Attenuate the Progression of Pancreatic Cancer Promoted by Porphyromonas Gingivalis in *K-ras^G12D^* Transgenic Mice

**DOI:** 10.3390/cancers12123522

**Published:** 2020-11-26

**Authors:** Shan-Ming Chen, Li-Jin Hsu, Hsiang-Lin Lee, Ching-Pin Lin, Szu-Wei Huang, Caucasus Jun-Lin Lai, Chia-Wei Lin, Wan-Ting Chen, Yu-Jen Chen, Yu-Chien Lin, Chi-Chieh Yang, Ming-Shiou Jan

**Affiliations:** 1Department of Pediatrics, Chung Shan Medical University Hospital, Taichung 40201, Taiwan; cshy228@csh.org.tw; 2Department of Pediatrics, School of Medicine, Chung Shan Medical University, Taichung 40201, Taiwan; 3Department of Medical Laboratory Science and Technology, Medical College, National Cheng Kung University, Tainan 70101, Taiwan; ljhsu@mail.ncku.edu.tw; 4Department of Surgery, Chung Shan Medical University Hospital, Taichung 40201, Taiwan; s31079@gmail.com; 5Institute of Medicine, Chung Shan Medical University, Taichung 40201, Taiwan; anitayen1971@yahoo.com.tw (C.-P.L.); caucasusgre@gmail.com (C.J.-L.L.); 6School of Medicine, Chung Shan Medical University, Taichung 40201, Taiwan; 7Division of Hepatology and Gastroenterology, Department of Internal Medicine, Chung Shan Medical University Hospital, Taichung 40201, Taiwan; 8Department of Post-Baccalaureate Veterinary Medicine, Asia University, Taichung 40201, Taiwan; holancat@asia.edu.tw; 9Department of Medical Laboratory and Biotechnology, Chung Shan Medical University, Taichung 40201, Taiwan; 10Institute of Biochemistry, Microbiology, and Immunology, Chung Shan Medical University, Taichung 40201, Taiwan; love05231227@gmail.com (C.-W.L.); dami811208@gmail.com (W.-T.C.); james99126@gmail.com (Y.-J.C.); wasitwo@hotmail.com (Y.-C.L.); 11Department of Hepatogastroenterology, Department of Internal Medicine, Show Chwan Memorial Hospital, Changhua 50093, Taiwan; 12Division of Allergy, Immunology and Rheumatology, Department of Internal Medicine, Chung Shan Medical University, Taichung 40201, Taiwan

**Keywords:** caerulein, *Lactobacillus*, pancreatic ductal adenocarcinoma, *Porphyromonas Gingivalis*, transforming growth factor-β

## Abstract

**Simple Summary:**

Pancreatic cancer is aggressive and lethal with a five year survival rate of only 5–9%. While the exact pathogenesis of pancreatic cancer is not fully understood, oral pathogens associated with periodontitis, such as *Porphyromonas gingivalis* (*P. gingivalis)*, are linked to the disease. The aim of our study was to investigate the causal association between exposure to *P. gingivalis* and subsequent carcinogenesis, and the potential modulatory effects of probiotics. We demonstrated that oral exposure to *P. gingivalis* can accelerate the development of pancreatic ductal adenocarcinoma in mouse models. In addition, the transforming growth factor-β (TGF-β) signaling pathway may be involved in the cancer-promoting effect of *P. gingivalis* and the suppressive effects of probiotics. Further understanding of the mechanisms of tumor-promoting or tumor-suppressing effects of TGF-β signaling may have potential as a treatment for pancreatic cancer.

**Abstract:**

Accumulating evidence suggests that there is a link between the host microbiome and pancreatic carcinogenesis, and that *Porphyromonas gingivalis (P. gingivalis)* increases the risk of developing pancreatic cancer. The aim of the current study was to clarify the role of *P. gingivalis* in the pathogenesis of pancreatic cancer and the potential immune modulatory effects of probiotics. The six-week-old LSL-K-rasG12D; Pdx-1-cre (KC) mice smeared P. gingivalis on the gums, causing pancreatic intraepithelial neoplasia (PanIN) after four weeks to be similar to the extent of lesions in untreated KC mice at 24 weeks. The oral inoculation of *P. gingivalis* of six-week-old *LSL-K-ras^G12D^*; *Pdx-1-cre* (KC) mice caused significantly pancreatic intraepithelial neoplasia (PanIN) after treatment four weeks is similar to the extent of lesions in untreated KC mice at 24 weeks. The pancreas weights of *P. gingivalis* plus probiotic-treated mice were significantly lower than the mice treated with *P. gingivalis* alone (*P* = 0.0028). The histological expressions of Snail-1, ZEB-1, collagen fibers, Galectin-3, and PD-L1 staining in the pancreas were also notably lower. In addition, probiotic administration reduced the histological expression of Smad3 and phosphorylated Smad3 in *P. gingivalis* treated KC mice. We demonstrated that oral exposure to *P. gingivalis* can accelerate the development of PanIN lesions. Probiotics are likely to have a beneficial effect by reducing cancer cell proliferation and viability, inhibiting PanIN progression, and cancer cell metastasis (Epithelial–mesenchymal transition, EMT). The transforming growth factor-β signaling pathway may be involved in the tumor suppressive effects of probiotics.

## 1. Introduction

Pancreatic cancer is one of the most aggressive and highly lethal malignancies. Even with recent advances in medicine, the overall five year survival rate is only 5–9% [1,2]. According to the GLOBOCAN 2018 estimates, it is the seventh most lethal cancer (4.5%) worldwide. Globally in 2018, 458,918 new cases were registered and there were 432,242 reported deaths [3]. This means that the mortality rate is nearly the same as the incidence rate. It is estimated that by 2030, pancreatic cancer will become the second leading cause of cancer-related deaths in the United States [4]. Although the exact pathogenesis of pancreatic cancer is not fully understood, some factors, such as smoking, alcohol abuse, obesity, diabetes, chronic pancreatitis, diet, and genetics have been identified as contributing factors [5].

The most important and prevalent type of malignant pancreatic neoplasm is pancreatic ductal adenocarcinoma (PDAC), which has a high risk of metastasis and accounts for approximately 90% of all pancreatic cancer cases [6]. Most PDAC cases (97%) have genetic alterations and four main driver genes have been identified [7]. A *K-ras* oncogenic mutation is present in up to 90% of all cases, including primary or metastatic tumors, and in pancreatic intraepithelial neoplasia (PanIN) lesions, which are a major subtype of PDAC precursor lesions [7]. Inactivation of tumor suppressor genes, including TP53, CDKN2A and SMAD4, also contributes to the progression of precursor lesions. Growing evidence suggests that there is a link between the host microbiome and PDAC. The proposed disease mechanism may be mediated by bacterial metabolites from the intestine or oral cavity, or bacterial translocation to the pancreas coupled with impaired pancreatic barrier function, which then causes the bacteria to colonize the pancreas and change its immune tolerance, thereby promoting the PDAC process [8]. Michaud et al. (2013) reported that people with high levels of antibodies against *Porphyromonas gingivalis* (*P. gingivalis*), a major periodontal pathogen, had twice the risk of pancreatic cancer compared with the control group (odds ratio (OR) = 2.14; 95% confidence interval (CI), 1.05–4.36) [9]. A population-based epidemiological study also revealed that carriage of *P. gingivalis* led to a significantly increased risk (60%) of developing pancreatic cancer (aOR = 1.60; 95% CI 1.15–2.22) [10]. Moreover, a recent study observed that the gut and pancreas microbiome in mouse and human PDAC was distinct and stage-specific, and that the endogenous microbiome may drive the progression of cancer by inducing intra-tumoral immune suppression. Conversely, targeting the microbiome distinctly protected against PDAC and improved the anti-tumor immune response [11]. These findings suggest that microbial targeted therapies may offer adjuvant benefits compared with standard therapies.

Using the *LSL-K-ras^G12D^*; *Pdx-1-cre* (KC) mouse model, the current study aimed to investigate the carcinogenic effect of *P. gingivalis* in PDAC development, and the potential immune modulatory effects of probiotic-based regimens.

## 2. Results

### 2.1. KC Mice Developed Pancreatomegaly and PanIN Lesions

The KC mouse model was used to experimentally study the disease processes. There was no significant difference in body weight between the non-treated wild type mice and the non-treated KC mice at 3, 6, 12, and 24 weeks (Figure 1A). We observed no statistically significant differences in the pancreas weights of the non-treated wild type and non-treated KC mice at 3, 6, and 12 weeks, but the pancreas weights of the non-treated KC mice were significantly increased at 24 weeks (0.4742 ± 0.5568 g) compared with the non-treated wild type mice (0.2467 ± 0.1967 g; *P* = 0.0023) (Figure 1B,C). A comparison of the non-treated wild type and non-treated KC mice pancreas histology by H and E staining is shown in Figure 1D. Non-treated wild type mice had normal tissues at 3, 6, 12, and 24 weeks. Non-treated KC mice developed pancreatic intraepithelial neoplasia (PanIN) lesions at 6, 12, and 24 weeks. Proliferating cell nuclear antigen (PCNA) immunostaining of the pancreas sections revealed that non-treated KC mice began to proliferate significantly at 12 weeks (Figure 1E).

### 2.2. P. gingivalis Accelerated the Development of PanIN Lesions

Figure 2A shows the time course for the mice treated with *P. gingivalis* to develop a mouse model of PanIN. The *P. gingivalis* treated KC mice showed PanIN lesions at 4 weeks after treatment. Similar changes were only observed in non-treated KC mice at 28 weeks (Figure 2B). A comparison of pancreas histology between caerulein-treated and *P. gingivalis-*treated six-week-old mice is shown in Figure 3A. The KC mice that were treated with caerulein or *P. gingivalis* showed PanIN lesions in their tissues. Analysis of cell proliferation by PCNA immuno-staining showed that caerulein or *P. gingivalis* induction increased the proliferation of pancreatic duct cells in KC mice at 4 weeks after treatment (Figure 3B). The wild type mice also showed mild proliferation of duct cells in the PCNA staining at four weeks after *P. gingivalis* induction (Figure 3B).

### 2.3. Probiotics Attenuate Pancreatic Carcinogenesis

Figure 4A shows the time course of *P. gingivalis*-treated mice with the co-administration of probiotics (GMNL 89:GMNL133 in 1:1 ratio). We observed that *P. gingivalis*-treated KC mice had a larger pancreas weight at four weeks after treatment compared with the non-treated KC mice. *P. gingivalis* plus probiotic-treated KC mice had a significantly lower pancreas weight compared with the *P. gingivalis*-treated KC mice who did not receive probiotics (*P* = 0.0028) (Figure 4B). The quantitative analysis of the number of normal pancreatic ducts and the number of PanIN lesions at different grade are shown in Figure 4C. Twelve week old KC mice that were treated with *P. gingivalis* and a combination of four week probiotics had significantly reduced PanIN-3 lesions compared with the group treated with *P. gingivalis* without probiotics (*P* = 0.0006). H and E staining of the pancreatic sections revealed that *P. gingivalis* plus probiotic-treated KC mice developed lower graded PanIN lesions at 12 weeks, compared with the mice that received *P. gingivalis* alone (Figure 4D). Notably, lower cell proliferation was also observed in the PCNA immuno-stained tissues (Figure 4E).

Furthermore, trichrome staining of the pancreatic sections revealed that *P. gingivalis* plus probiotic-treated KC mice developed less extensive perilobular fibrosis, compared with the mice treated with *P. gingivalis* alone (Figure 5A). There was lower expression of epithelial-mesenchymal transition (EMT)-related markers (Snail-1 and ZEB-1) in the *P. gingivalis* plus probiotic-treated KC mice as shown in Figure 5B,C. PD-L1-positive and Galectin-3-positive staining around the pancreas duct and its surroundings were significantly reduced in the *P. gingivalis* plus probiotic-treated KC mice, compared with the non-treated KC mice and the KC mice treated with *P. gingivalis* alone (Figure 6A,B).

### 2.4. Probiotics Inhibit the Activation of the Transforming Growth Factor (TGF)-β Signaling Pathway

In KC mice administered oral *P. gingivalis*, the immune expression of Smad3 and TGF-β downstream molecules, was significantly increased in the cytoplasm of pancreatic cells. In addition, increased expression of phosphorylated Smad3 (pSmad3) was also observed in the nucleus, along with a lower expression level of Smad7 (Figure 7A,B). KC mice that were treated with *P. gingivalis* and probiotics administration had reduced expression of Smad3 and pSmad3, but the expression of Smad7 was inversely increased.

### 2.5. The Lysates of Probiotics Significantly Decreased Pancreatic Cancer Cell Viability

To further illustrate the effect of GMNL-89 and GMNL-133 on the sensitivity of cancer cells in vitro, the cell viability was measured via MTT assay. The cell viability in the standard chemotherapeutic agent gemcitabine (GCB) combined with probiotic lysate (GMNL-89 + GMNL-133) group was markedly decreased compared to the standard chemotherapy (*P* < 0.001) (Figure 8A). Probiotic lysate (GMNL-89 + GMNL-133) alone also showed the effect of reducing the viability of cancer cells compared to the control group (*P* < 0.001) (Figure 8B).

## 3. Discussion

There is growing evidence to suggest that chronic inflammation is a cofactor of carcinogenesis and is involved in the diverse steps leading to it, including cell transformation, proliferation, invasion, angiogenesis, and metastasis [12]. Between 15% and 20% of all cancer deaths worldwide can be attributed to infectious agents and inflammatory responses, and these are the main components of chronic inflammation [13]. An increased risk of developing cancer is associated with chronic inflammation triggered by chemical and physical agents, autoimmune diseases, and inflammatory reactions of uncertain etiology [12]. However, not all chronic inflammation promotes carcinogenesis. The pro-inflammatory tumor microenvironment (TME), the cellular environment surrounding tumor cells, plays a critical role in the subsequent evolution of tumors. The TME is composed of immune cells, stromal cells, blood vessels, signaling molecules and the extracellular matrix (ECM) [14,15]. Conversely, current research has observed that a healthy microenvironment helps to prevent the occurrence and invasion of tumors [15]. Over the past few decades, understanding of the TME has influenced our understanding of the evolution of cancer, and treatment strategies have also gradually changed from a tumor centered approach to management of the tumor ecosystem. The immune system, an important determinant of the TME, plays an essential role in host immune surveillance and the elimination of nascent cancer cells [16]. In addition, the dual roles of inflammatory mediators such as growth factors, chemokines, and cytokines, can suppress or promote inflammation-induced cancer [17]. PDAC is a typical example of an inflammation-driven cancer. Patients with chronic pancreatitis have a 13-fold increased risk of developing PDAC [18,19]. Accumulating evidence suggests there is an association between pancreatic carcinogenesis and specific bacterial infections; the microbiome may affect tumor development and modulate the TME, as well as influencing the interplay between inflammation and tumor progression [19]. To investigate the potential role of oral bacterial pathogens and probiotics in pancreatic carcinogenesis, we conducted this animal trial using K-ras-driven mouse models of PDAC.

*P. gingivalis* is a gram-negative oral anaerobic bacterium and one of over 500 bacterial species found in the oral cavity, which is related to the pathogenesis of periodontitis. A bacterial complex called “red complex” that is composed of *P. gingivalis*, *Treponema denticola*, and *Tannerella forsythia* is believed to be closely associated with advanced periodontal disease [20]. Studies to date have shown that there is a correlation between *P. gingivalis* and digestive system cancers, including oral cancer, esophageal cancer, and pancreatic cancer [21,22]. Li et al. reviewed epidemiological studies since 2003 and showed that periodontal disease increases the risk of pancreatic cancer [23]. This provides some indirect evidence for the association between *P. gingivalis* and pancreatic cancer. Research using plasma antibody detection and oral mouth swabs for microbiota composition further support this association between *P. gingivalis* and pancreatic cancer [9,10]. However, there is still a lack of direct evidence for the causal association between exposure to *P. gingivalis* and pancreatic inflammation and subsequent carcinogenesis.

As with many previous studies, in the current study the KC mice were more prone to develop PanIN lesions compared with the wild type mice. Caerulein, a cholecystokinin analog, is known to induce acute pancreatitis in mouse models. The current study showed that *P. gingivalis* also induces PanIN lesions in KC mice just like caerulein. Even in *P. gingivalis*-treated wild type mice, the proliferation of pancreatic duct cells was observed at six weeks. The results of the current animal trial support the role of *P. gingivalis* in the oral cavity as a direct promoter or co-promotor that causes inflammation of the pancreas and even carcinogenesis. Previous animal experiments on oral cancer have provided a preliminary understanding of the mechanism of EMT and oral squamous cell carcinoma (OSCC) progression following *P. gingivalis* infection [22]. After infection with *P. gingivalis*, phosphatidylinositol-3-kinase/protein kinase B (PI3K/Akt) signaling is increased to inhibit intrinsic apoptosis, thereby promoting the survival and proliferation of epithelial cells. *P. gingivalis* also blocks extracellular ATP/P2 × 7 danger signaling and protects itself and host epithelial cells from the reactive oxygen species produced by damaged mitochondria and NAPDH Oxidase 2 (NOX2). In addition, *P. gingivalis* promotes EMT by inactivating Glycogen synthase kinase 3 beta (GSK3-β) and promoting the conversion of E-cadherin to vimentin. It also increases the expression of cancer stem cell markers, such as CD44 and CD133 [22]. *P. gingivalis* is able to block the p53 tumor suppressor to maintain the survival and proliferation phenotype in cancer cells. *P. gingivalis* induces the production of promatrix metalloproteinase-9 (pro-MMP9). Invasive OSCC cell lines are promoted through activation of this proenzyme to MMP9 by gingipains (key virulence factors). In addition, *P. gingivalis* regulates the immune environment via increased expression of B7-H1 and B7-DC receptors, which lead to T-cell anergy and activated T-cell apoptosis [22]. However, whether *P. gingivalis* migrates to the pancreas to cause direct damage or has a long-distance effect on pancreatic carcinogenesis, and how it affects cancer progression, still needs further research.

Growing preclinical evidence suggests that microbes are associated with PDAC susceptibility, initiation, progression, and therapeutic sensitivity [24]. Thomas et al. reported that microbiota-depleted mice showed a decreased proportion of poorly differentiated neoplasm compared with microbiota-intact mice. They also observed that malignant tissue in the pancreas harbored microbiota and supported the role of the host microbiota in the long-distance effects of PDAC progression [25]. Pushalkar et al. found that the distinct bacterial dysbiosis associated with PDAC led to innate and adaptive immunosuppression. They suggested that immune suppression of the PDAC microbiome and macrophage programming is TLR-dependent, and that the microbiome acts as a potent modulator of the inflammatory TME [11]. Sethi et al. reported that when the gut microbiome is depleted by oral antibiotics it is able to increase the number of Th1 (IFN-γ^+^CD4^+^CD3^+^) and Tc1 (IFN-γ^+^CD8^+^CD3^+^) cells in the TME, increase the number of anti-tumor IFN-γ-secreting T cells (IFN-γ^+^CD3^+^), and decrease the number of pro-tumor IL17a (IL17a^+^CD3^+^) and IL10 (IL10^+^CD4^+^CD3^+^) secreting cells, which results in a reduced pancreatic tumor burden in mice [26]. A recent study reported that the intestinal microbiome of patients with PDAC accounts for approximately 25% of the microbiome in their pancreatic tumors, but it is not found in adjacent normal pancreatic tissues. Fecal microbial transplantation from long-term survivors exhibited increased levels of CD8+ T cells and activated CD8+ T cells, IFN-γ and IL-2, and improved survival in a mouse model [27]. These findings indicate that the gut microbiota is able to colonize the pancreatic tumor tissue and modulate tumor progression; thus, manipulation of the intestinal microbiota may have potential as a novel immunotherapy strategy.

Previous animal experiments indicate that oral administration of *P. gingivalis* can cause changes in the intestinal flora and increase the risk of various diseases by changing its metabolic profiles [28,29]. Related studies also show that *Lactobacillu*s and *Bifidobacterium* may have the ability to inhibit *P. gingivalis* [30,31]. A review study pointed out that there is a correlation between pancreatic cancer and *P. gingivalis*, and chronic pancreatitis has lower intestinal *Lactobacillu*s and *Bifidobacterium* [32]. Liu et al. reported the effect of *Lactobacillus reuteri* on the intestinal flora of healthy mice [33]. In addition to the proliferation of beneficial intestinal flora, it also increases the levels of tryptophan metabolites and purine nucleoside adenosine to improve tolerance to inflammatory stimuli. Here we infer that the administration of *Lactobacillu*s may be a strategy to improve pancreatic inflammation and prevent pancreatic cancer. Supplementation with *Lactobacillus reuteri* GMNL-89, the first probiotic used in the current study, has been previously shown to increase antioxidant activity and reduce IL-6 levels in animal models [34]. Previous evidence has indicated that IL-6 has pro-tumorigenic activity in the progression of PDAC, and high serum IL-6 levels are associated with a worse prognosis [35]. Previous research has shown that Th2 responses exhibit tumor-promoting effects and are associated with a poor prognosis in PDAC [36]. *Lactobacillus paracasei* GMNL-133, the second probiotic used within the present study, has been shown to be beneficial in pediatric asthma, which is thought to be a Th2-driven disease [37]. This probiotic has been shown to inhibit Th2 cytokine production and modulate the Th1/Th2 immune balance by increasing IFN-γ levels. These findings indicate that the *P. gingivalis* and probiotic-treated-mice may have less pancreatic inflammation. There was light staining for the Snail-1 and ZEB-1 transcription factors in the co-administration group, suggesting that the probiotics inhibited the start of the EMT process. This further indicates that probiotics could play a role in reducing invasion and metastasis in PDAC. Galectin-3 is considered to be a metastasis-inducing protein that is expressed by tumor and inflammatory cells. It binds with oligosaccharides on the extracellular domain of integrin αvβ3 in a multimeric manner and also mediates *K-ras* aggregation on the plasma membrane and enhances *K-ras* activity [38]. A recent study has shown that galectin-3 mediates tumor cell-stromal interactions in pancreatic cancer [39]. In the current study, we observed that galectin-3 staining of the pancreatic tissue in probiotic treated mice was reduced compared with the mice not treated with probiotics, indicating that probiotics have the potential to suppress malignant cellular transformation and metastasis of cancer cells. In addition to immune regulation, probiotics can also eliminate harmful pathogens through non-immune mechanisms such as lowering the pH of the lumen, competing for nutrients, and producing bacteriocin-like substances [40]. The previous report indicated that in patients with atopic dermatitis after taking *Lactobacillus paracasei* GMNL-133 for three months, the number of *Bifidobacterium* in stool will be significantly higher than that in the placebo group [41]. *Bifidobacterium* has been shown to improve therapeutic activity of anti-PD-L1 treatment in the report of animal experiments [42]. In our study, we have also observed the inhibition of probiotic strains on pancreatic cancer cells in vitro. The use of *Lactobacillus reuteri* GMNL-89 plus *Lactobacillus paracasei* GMNL-133 with the standard chemotherapeutic agent GCB to inhibit cancer cells is better than the use of single bacteria or only chemotherapy. Even if *Lactobacillus reuteri* GMNL-89 is used alone without the GCB, it still has the effect of reducing the cancer cells. If it is used in combination with *Lactobacillus paracasei* GMNL-133, it will have a better inhibitory effect.

It is now known that the PD-1/PD-L1 interaction activates downstream signals to inhibit T cell activation, resulting in the occurrence of tumor immune escape. Previous studies have shown that PD-L1 can be expressed in PDAC, and its overexpression is associated with drug resistance and poor prognosis [43]. Our results show that probiotic treatment to inhibit the expression of PD-L1 could be used as an adjuvant treatment with immune checkpoint inhibitors to improve the efficacy of anti-cancer drugs and the patient’s overall prognosis. In addition, prior evidence has indicated that TGF-β and its downstream signaling molecules play a critical role in pancreatic carcinogenesis [44]. The overexpression of Smad3 in PDAC correlates with EMT induction and metastasis, and poor prognosis [45,46]. Smad7 is a critical negative regulator of TGF-β signaling and may play a dual role as a proliferation promoting factor or a metastatic suppressor in different human cancers [47,48]. A previous study reported that low levels of Smad7 expression in pancreatic cancer are closely related to lymph node metastasis and poor prognosis [49]. The current study indicates that *P. gingivalis* promotes activation of the TGF-β signaling pathway in pancreatic ductal cells and that the TGF-β signaling pathway may be involved in the inhibitory effect of probiotics on the progression of PDAC. Further studies are warranted to clarify the exact mechanism of action of *P. gingivalis* and to evaluate the role of probiotics in pancreatic cancer therapy.

The present study offers some clinical implications for the management of pancreatic cancer. We have demonstrated that *P. gingivalis* is directly associated with the development of PDAC in animal experiments. Prevention or early diagnosis of periodontal disease may have a significant impact on the onset of disease and treatment in people with an established risk of pancreatic cancer. A second important implication derived from our results is that the addition of probiotics as an adjuvant should be considered as a potential therapeutic strategy, especially given the current limited effectiveness of traditional chemotherapy. Moreover, this adjuvant treatment is worthy of further clinical research to determine its potential ability to manipulate the human intestinal flora, and its impact on the TME. It is important to also develop more accurate biomarkers and better non-invasive early screening strategies. Nevertheless, there were several limitations to the present study. First, the current mouse PDAC model has not been able to completely simulate the pathological process of human PDAC. In humans, the majority of pancreatic cancers present as a single tumor, while in Genetically Engineered Mouse (GEM) models, it shows with a multifocal growth and spread pattern [50]. Many important genes related to human pancreatic cancer are not expressed in the mouse model [50]. Compared with the complexity and duration of human tumor development, the mouse model may have some simplifications. These differences may cause the mouse PDAC model to be different from human PDAC in biology and treatment response. Second, the cross-talk between the intestinal microbiota and its host is considered host-specific, and as such not all results observed in mouse models can be replicated in humans. Even experiments in different mouse models will show different results. It is necessary to further analyze the impact of *P. gingivalis* infection on the intestinal microbiome and host-microbe interactions to understand the relationship between cancer progression and the potential anti-cancer effects of probiotics in humans.

## 4. Materials and Methods

### 4.1. P. gingivalis Culture

*P. gingivalis* (ATCC^®^ 33277™) was obtained from the Bioresource Collection and Research Center, Taiwan. The bacteria were cultured in Tryptic soy broth media (TSB) containing 0.5% yeast extract (Difco), 1 μg/mL vitamin K, 5 μg/mL hemin, and a 0.05% L-cysteine reducing agent. All reagents were purchased from Sigma-Aldrich (St. Louis, MO, USA), with the exception of the TSB and yeast extract which were purchased from Difco Laboratories (Detroit, MI, USA). The *P. gingivalis* were cultured at 37°C under anaerobic condition with 5% H_2_, 10% CO_2_, and 85% N_2_. *P. gingivalis* bacteria and cultured broth were separated by centrifugation. The bacteria were then washed twice with sterile phosphate-buffered saline. The *P. gingivalis*-cultured broth was filtered using a 0.22 µm sterile filter. Afterwards, the aliquot, bacteria, and cultured broth were stored at −80 °C prior to use.

### 4.2. Transgenic Pancreatic Cancer Animal Model

B6.129-*K-ras^tm4TYj^* and B6.FVB-Tg(Pdx-1-cre)1Tuv mice were imported from the National Cancer Institute Mouse Repository, USA. *LSL-K-ras^G12D+/-^*-*Pdx-1-cre^+/-^* dual-gene transgenic mice were bred in the Animal Center at Chung Shan Medical University, in Taiwan. All the breeding and animal experiments were approved by the Institutional Animal Care and Use Committee of the Animal Center at Chung Shan Medical University (IACUC nos. 1461 and 1775) and were carried out in accordance with the Committee’s policies. Animals were bred in positive pressure ventilated cages supplied with sterilized reverse osmosis (RO) water and standard feed. Light was provided for 12 h a day, and the temperature and humidity were maintained at 22 to 24 °C and 60%, respectively.

### 4.3. Oral Smear of P. gingivalis

Mice were allowed to freely drink sterile potable water containing antibiotics (0.80 mg/mL sulfamethoxazole and 0.16 mg/mL trimethoprim) for ten days. The water was changed every five days. On day 11, the water was changed to sterile water without antibiotics for three days to allow for the removal of residual antibiotics by metabolism. Then, 0.2 mL saline containing 1 × 10^9^ viable *P. gingivalis*, 0.2% L-cysteine, and 2% carboxymethylcellulose was oral cavity-smeared on the experimental group (*P. gingivalis-*treated mice) once every two days, for a total of three smears. The control group (non-treated mice) was given three 0.2 mL smears of a similar solvent (*P. gingivalis*-free).

### 4.4. Probiotic Administration

*Lactobacillus reuteri* GMNL-89 and *Lactobacillus paracasei* GMNL-133 were prepared and provided by GenMont Biotech Inc. (Tainan, Taiwan). The probiotic for the mice contained GMNL-89:GMNL-133 in a 1:1 ratio (0.82 × 10^7^ CFU/0.02kg:0.82 × 10^7^ CFU/0.02kg). The lyophilized live probiotic powders were resuspended in sterile Milli-Q water and then mixed well by gentle vortexing. This was then administered to the mice (1.64 × 10^7^ CFU/0.2 mL/mice) orally through a stainless-steel gavage needle for 5 consecutive days per week for four weeks.

### 4.5. Histology and Immunohistochemistry

Pancreatic tissues were fixed by formaldehyde and embedded in paraffin. The cancer stage was determined by hematoxylin/eosin Y (H/E) staining of the tissue in accordance with current definitions. Immunohistochemistry of the embedded tissues followed the standard procedure, including tissue slicing, dewaxing, rehydration, and antibody staining. Proliferating cell nuclear antigen (PCNA) is a catalyzer for DNA polymerase delta and is essential for cell replication. PCNA expression is usually used as a marker of cell proliferation based on the characteristics of PCNA protein that remain longer in the G1/S phase. Antibodies used included anti-PCNA (Abcam #ab92552), anti-Snail-1 (Santa Cruz, sc-28199), anti-ZEB-1 (Santa Cruz, sc-81428), anti-PD-L1 (Proteintech #17952-AP), anti-Galectin-3 (Novus #NB100-91778), anti-Smad3 (Novus #NB100-56479), anti-phospho-Smad3 (T179) (Abcam#74062), and anti-Smad7 (R and D system #MAB2029). The primary antibodies binding to the tissue sections were visualized using BOND Polymer Refine Detection (Leica Biosystems, DS9800). Tissue imaging and analysis were quantified by a TissueFAXS^200^ (TissueGnostics, GmbH) flow analyzer. Trichrome staining was performed using a Trichrome Stain kit (Modified Masson’s) (Scy Tek Lab Inc.).

### 4.6. Histologic Evaluation of Pre-Cancer Lesions (PanINs)

The formalin-fixed paraffin-embedded pancreatic tissue was sectioned and stained with hematoxylin and eosin, and the histological evaluation of each pancreas section was performed by a pathologist. To quantify the progression of PanIN lesions, we determined the total number of ductal lesions and their grade. According to the naming and classification system of pancreatic duct lesions described in reference [51], each counted cluster is classified into normal ducts, PanIN-1A, PanIN-1B, PanIN-2, and PanIN-3.

### 4.7. Cell Viability

Two thousand five hundred BxPC-3 cells (ATCC® CRL-1687™) were seeded in 96 well-plates at 37 °C overnight. Cells were treated with probiotic lysates of GMNL-89 (25 µg/mL; 6.41 × 10^8^ CFU/mL) or GMNL-133 (25 µg/mL; 7.76 × 10^8^ CFU/mL) in the presence or absence of gemcitabine for 0, 1, 2, 4, and 6 days. Cell viability was assessed by 3-(4,5-dimethylthiazol-2yl)-2,5-diphenyltetrazolium bromide (MTT) assay. Gemcitabine (dFdC, 2′, 2′-difluorodeoxycytidine) (Eli Lilly and Company, Indianapolis, IN, USA) is a fluorinated analog of deoxycytidine, which is the main chemotherapeutic drug for treating pancreatic cancers [52].

### 4.8. Statistical Analysis

All statistical analysis was performed using SPSS software version 10.1.3C (SPSS Inc., Chicago, IL, USA) for Windows. Comparison of the means of two independent samples was performed by Student’s *t-*test. A *P-*value of less than 0.05 was considered to indicate a statistically significant difference between groups.

## 5. Conclusions

We demonstrated that oral exposure to *P. gingivalis* can accelerate PanIN lesions in KC mice with a severity similar to that caused by caerulein. This clearly shows that *P. gingivalis* plays a direct role in the deterioration of PDAC. Our findings revealed that *Lactobacillus* had beneficial effects and could reduce the number and grades of PanIN lesions and the growth and survival of pancreatic cancer cells and inhibit EMT (suppress malignant cell transformation and metastasis) in cancer cells. The TGF-β signaling pathway may be involved in the cancer-promoting effect of *P. gingivalis* and suppressive effects of probiotics. The inhibitory effect of probiotics on PD-L1 expression may be considered as a promising adjuvant treatment to immune checkpoint inhibitors for selected patients with pancreatic cancer in the future.

## Figures and Tables

**Figure 1 cancers-12-03522-f001:**
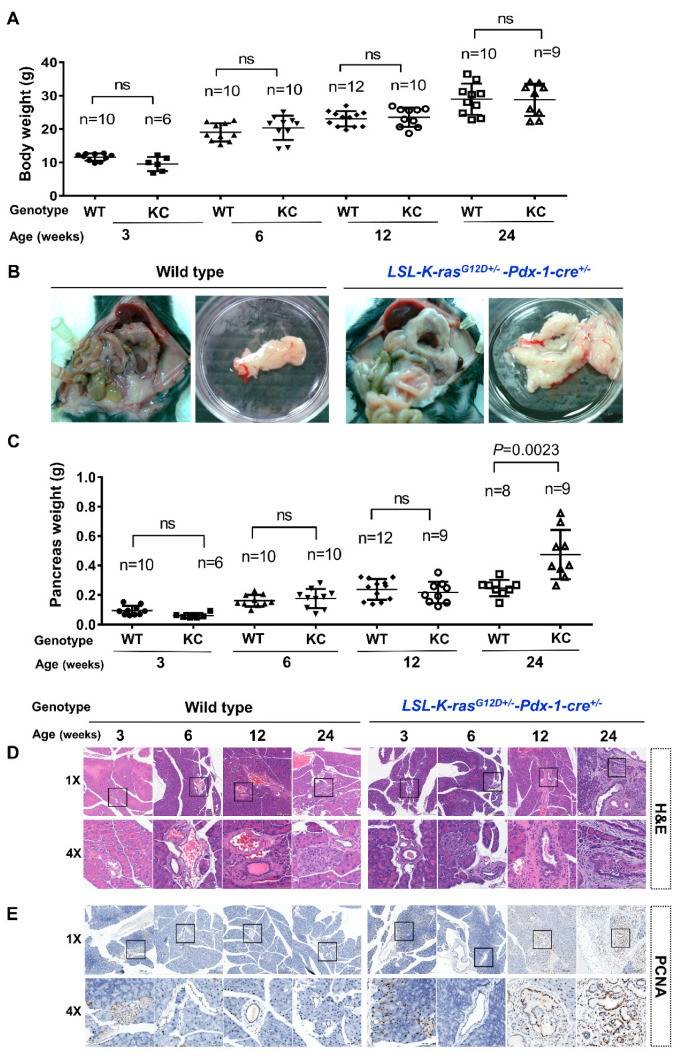
Accelerated pancreatic ductal adenocarcinoma (PDAC) progression in a transgenic *LSL-K-ras^G12D+/-^*-*Pdx-1-cre*^+/-^ mouse model. (**A**) No significant difference was observed between the body weights of non-treated KC mice and non-treated wild-type mice at 3, 6, 12, and 24 weeks. (**B**) Comparison of pancreas size for KC mice with age- and sex-matched wild type mice at 24 weeks. (**C**) KC mice showed significantly increased pancreas weights (*p* = 0.0023) at 24 weeks. (**D**) Representative H and E staining images of the mouse pancreas at 3, 6, 12, and 24 weeks. KC mice developed pancreatic intraepithelial neoplasia (PanIN) lesions at 6, 12, and 24 weeks (scale bar, 200 µm). (**E**) Representative images of proliferating cell nuclear antigen (PCNA) staining of the mouse pancreas, indicating a significant increase in cell proliferation at 12 weeks in KC mice (Figure 1E) (scale bar, 50 µm). WT, wild type; KC, *LSL-K-ras^G12D+/-^*-*Pdx-1-cre*^+/-^.

**Figure 2 cancers-12-03522-f002:**
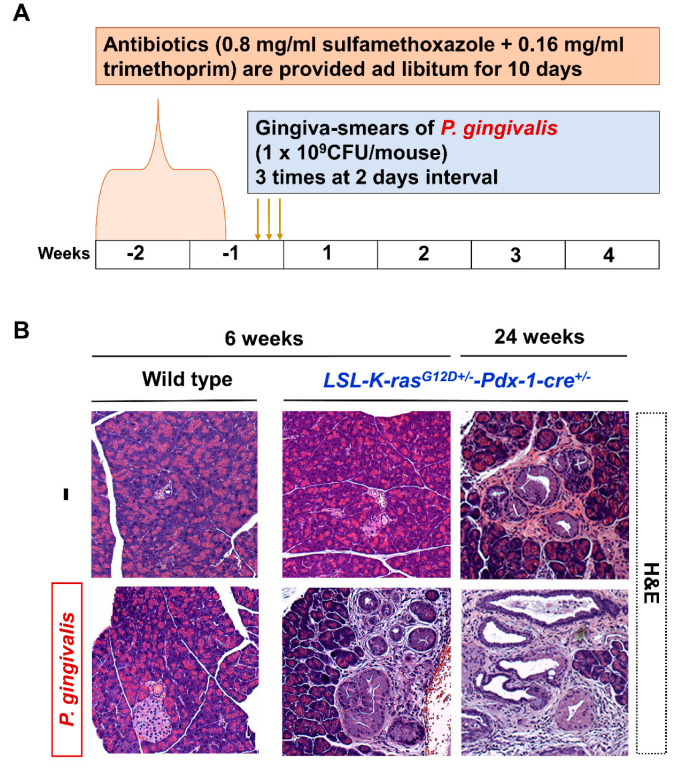
(**A**) Schematic diagram illustrating the time course of intraoral inoculation of *P. gingivalis* in wild type and KC mice. (**B**) Representative H and E staining images of mouse pancreas in indicated ages of wild type and KC mice either sham-operated or oral inoculation with *P. gingivalis*. Six-week-old KC mice treated with *P. gingivalis* significantly developed PanIN lesions at 10 weeks old, and non-treated 6-week-old KC mice developed few PanIN lesions. More severe of lesions found in pancreas of 24-week-old KC mice oral inoculation with *P. gingivalis*. Exposure to *P. gingivalis* significantly accelerated the progression of PanIN lesions. PanIN, pancreatic intraepithelial neoplasia.

**Figure 3 cancers-12-03522-f003:**
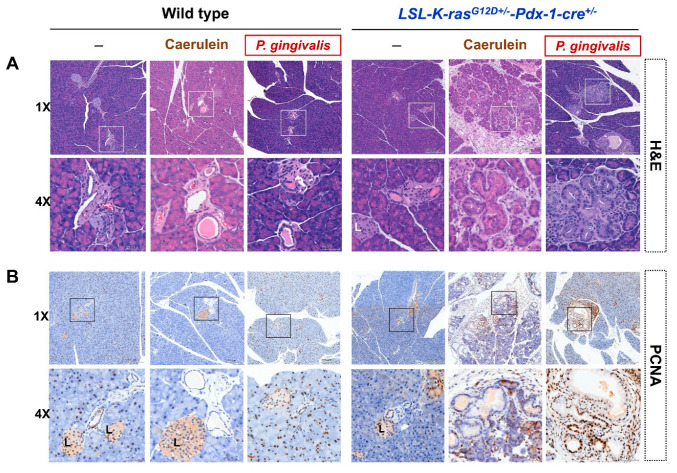
Representative images of pancreatic histology in non-treated mice, caerulein-treated mice, and *P. gingivalis-*treated mice at four weeks after treatment. (**A**) H and E staining (upper scale bar, 200 μm; lower scale bar, 50 μm). (**B**) PCNA staining (upper scale bar, 200 μm; lower scale bar, 50 μm). L, langerhans islets; PCNA, proliferating cell nuclear antigen.

**Figure 4 cancers-12-03522-f004:**
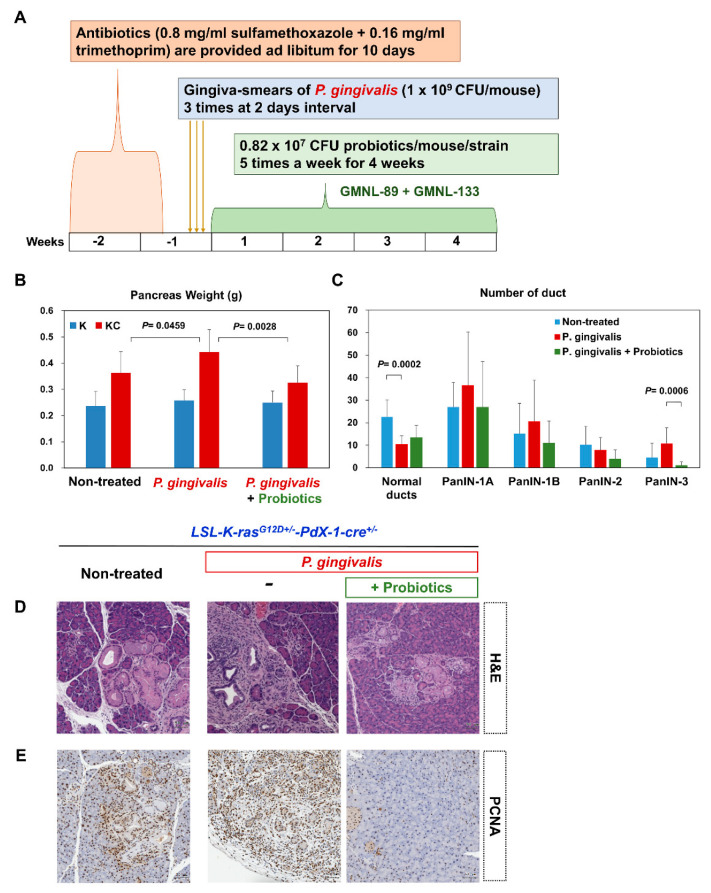
(**A**) Schematic diagram illustrating the time course of *P. gingivalis*-treated KC mice with the co-administration of probiotics (GMNL 89:GMNL133 in 1:1 ratio). (**B**) Comparison of pancreas weight in non-treated, *P. gingivalis*-treated, and *P. gingivalis* plus probiotic-treated mice. The pancreas weight of the *P. gingivalis* plus probiotic-treated mice was significantly reduced compared with the *P. gingivalis*-treated mice (*P* = 0.0028). (**C**) Number of normal ducts and PanIN lesions were counted and compared between the non-treated, *P. gingivalis*-treated, and *P. gingivalis* plus probiotic-treated mice at four weeks after treatment. The number of normal ducts for the *P. gingivalis* mice was significantly reduced compared with the non-treated mice (*P* = 0.0002). The number of PanIN-3 lesions for the *P. gingivalis* plus probiotic-treated mice was significantly reduced compared with the *P. gingivalis*-treated mice (*P* = 0.0006). Bar graph shows the mean number of duct ± standard deviation (SD) of a mouse pancreatic tissue section at 100x magnification, n = 10. (**D**) Representative H and E staining of pancreatic tissue in non-treated KC mice, *P. gingivalis*-treated mice, and *P. gingivalis* plus probiotic-treated mice at 12 weeks (scale bar, 50 µm). (**E**) Representative PCNA staining of pancreatic tissue in non-treated KC mice, *P. gingivalis*-treated mice, and *P. gingivalis* plus probiotic-treated mice at 12 weeks (scale bar, 50 µm). K, *LSL-K-ras^G12D+/^*^-^; KC, *LSL-K-ras^G12D+/-^*-*Pdx-1-cre*^+/-^; PCNA, proliferating cell nuclear antigen.

**Figure 5 cancers-12-03522-f005:**
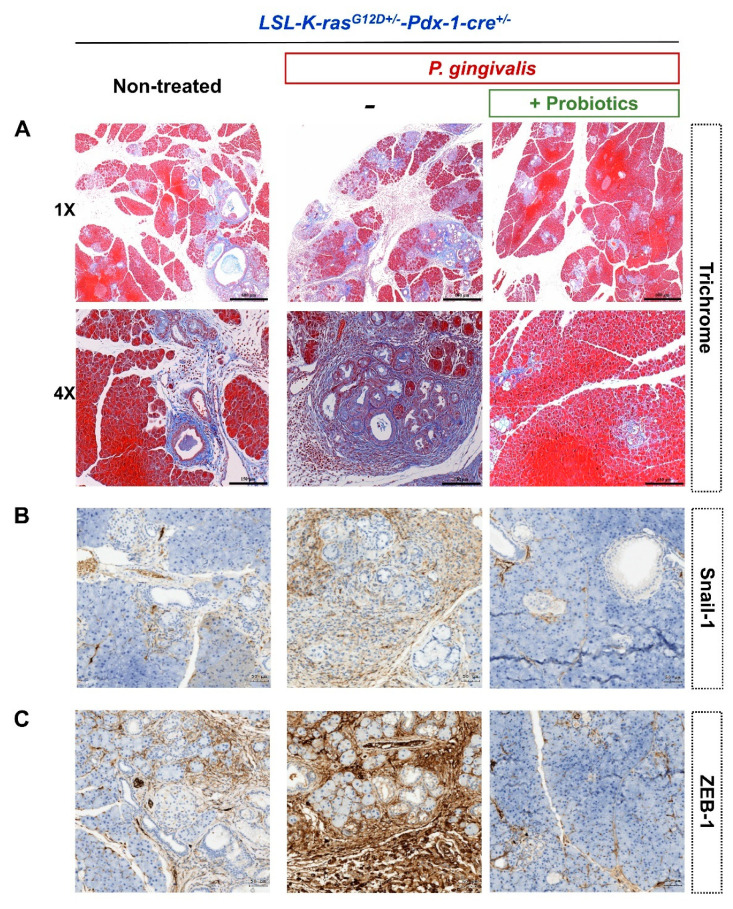
Representative images of pancreatic histology by various staining methods in non-treated KC mice, *P. gingivalis*-treated KC mice, and *P. gingivalis* plus probiotic-treated KC mice at four weeks after treatment (GMNL 89: GMNL133 in 1:1 ratio). (**A**) trichrome staining (upper scale bar, 400 μm; lower scale bar, 150 μm). (**B**) Snail-1 staining (scale bar, 50 µm). (**C**) ZEB-1 staining (scale bar, 50 µm). KC, *LSL-K-ras^G12D+/-^-Pdx-1-cre*^+/-^; ZEB-1, Zinc finger E-box-binding homeobox 1.

**Figure 6 cancers-12-03522-f006:**
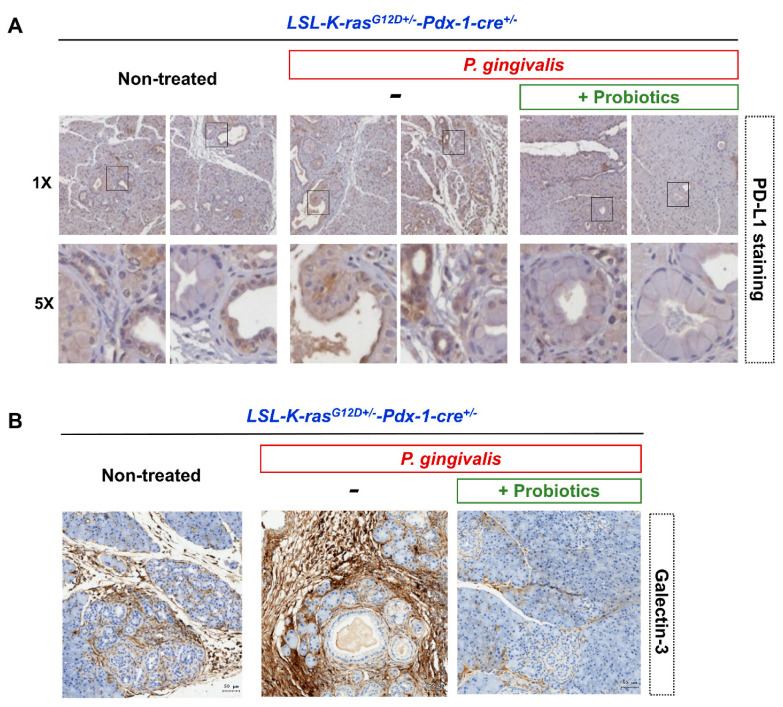
Representative images of immunomodulatory proteins in pancreatic tissues by immunohistochemistry analysis in non-treated KC mice, *P. gingivalis*-treated KC mice, and *P. gingivalis* plus probiotic-treated KC mice (GMNL 89: GMNL133 in 1:1 ratio). (**A**) PD-L1 staining. (**B**) Galectin-3 staining (scale bar, 50 µm). KC, *LSL-K-ras^G12D+/-^*-*Pdx-1-cre*^+/-^; PD-L1, programmed death-ligand 1.

**Figure 7 cancers-12-03522-f007:**
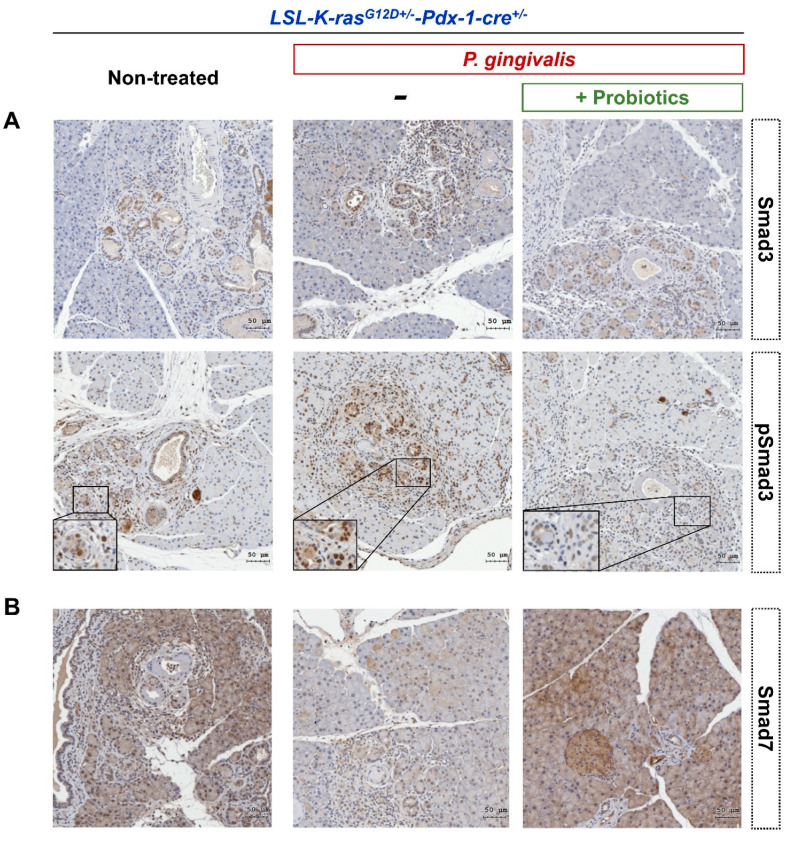
Representative expression of TGF-β signaling pathway-related proteins in pancreatic tissues by immunohistochemistry analysis in non-treated KC mice, *P. gingivalis*-treated KC mice, and *P. gingivalis* plus probiotic-treated KC mice (GMNL 89: GMNL133 in 1:1 ratio). (**A**) Smad3 and phospho-Smad3 staining. (**B**) Smad7 staining (scale bar, 50 µm). KC, *LSL-K-ras^G12D+/-^*-*Pdx-1-cre*^+/-^.

**Figure 8 cancers-12-03522-f008:**
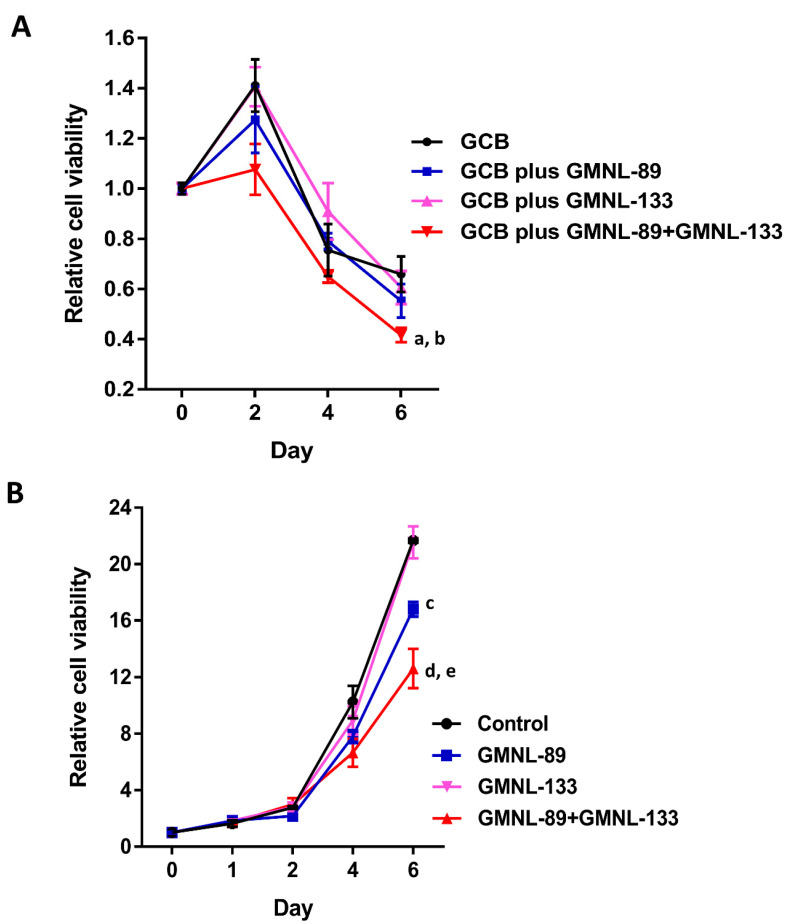
The relative cell viability of BXPC-3 pancreatic cancer cells in different probiotic lysate treatment (25 µg/mL) at different time points was measured by the MTT assay (OD570nm). The value of relative cell viability represents the fold change compared to baseline (Day 0) of untreated cell numbers. (**A**) In the presence of standard chemotherapeutic agent GCB ^a^
*P* < 0.001 differences between GCB and GCB with GMNL-89+GMNL-133. ^b^
*P* < 0.01 differences between GCB and GCB with GMNL-89. (**B**) In the absence of GCB ^c^
*P* < 0.001 differences between control and GMNL-89. ^d^
*P* < 0.001 differences between control and GMNL-89+GMNL-133. ^e^
*P* < 0.01 differences between GMNL-89 and GMNL-89+GMNL-133. GCB, gemcitabine.

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
