# Peer review of "Lactobacillus Attenuate the Progression of Pancreatic Cancer Promoted by Porphyromonas Gingivalis in K-rasG12D Transgenic Mice"

_cancers, 2020, doi:10.3390/cancers12123522_

Round 1

Reviewer 1 Report

Chen and colleagues sumitted an interesting research addressing the role of oral pathogen P.gingivalis on pathogenesis of pancreatic cancer, which can be targeted with lactobacillus spp. The authors also provide histochemical analyses of TGFbeta, EMT and PDL1 pathway activities to infer the underlying mechanism. The manuscript is clearly written, and addressed a novel perspective of microbiota-cancer interaction.

The key finding that administration of P. gingivalis promotes spontaneous pancreatic cancer formation in tissue-specific KRAS G12D background is very interesting. The authors put in reasonable amount of efforts to characterize the phenomenon unequivocally. It will be more convincing if the authors provide more supportive evidences of the possible cellular/tissue level mechanism of this phenomenon, and include quantitative analyses of IHC figures.

The fact that sequential application of lactobacillus antagonizes P gingivalis virulence indirectly indicate that the overall bacterial load is not a cause for PDAC formation in this model. Some data on plasma or local immune profiles along the study course may help clarify if modulation of host immune activity is a critical component. Composition of tumor infiltrating leucocytes will be especially useful to demonstrate if P gingivalis activates a regulatory immune microenvironment to allow progression of PIN to PDAC.

Finally, the authors probably want to discuss what is the virulence mechanism for P gingivalis, which spends most of the life cycle as an intracellular microbe, to promote PDAC formation from PIN. Although experimental proof of such mechanism is beyond the scope of this paper, this will provide an unique angle to hint on a novel perspective of carcinogenesis of this deadly disease and possibly evoke new intervention strategy.

Author Response

Samuel C. Mok, Ph.D.,

Editor-in-Chief

Cancers

Oct 29, 2020

Dear Prof. Mok,

Thank you very much for the helpful comments on our article (cancers-975891) entitled “Lactobacillus attenuate the progression of pancreatic cancer promoted by Porphyromonas gingivalis in K-rasG12D transgenic mice”. We acknowledge the reviewer’s comments, which are valuable in improving the quality of our manuscript. We have revised our manuscript accordingly and highlighted the changes in blue. In addition, we respond to the editorial board's comments point-by-point as listed below:

Reviewer 1

Point 1. Chen and colleagues sumitted an interesting research addressing the role of oral pathogen P. gingivalis on pathogenesis of pancreatic cancer, which can be targeted with lactobacillus spp. The authors also provide histochemical analyses of TGFbeta, EMT and PDL1 pathway activities to infer the underlying mechanism. The manuscript is clearly written, and addressed a novel perspective of microbiota-cancer interaction.

R1: The aim of the present study was to explore the carcinogenic role of P. gingivalis in pancreatic cancer and the potential effects of probiotic-based regimens. Thank you for your time and effort to enhance the insight and integrity of our study.

Point 2. The key finding that administration of P. gingivalis promotes spontaneous pancreatic cancer formation in tissue-specific KRAS G12D background is very interesting. The authors put in reasonable amount of efforts to characterize the phenomenon unequivocally. It will be more convincing if the authors provide more supportive evidences of the possible cellular/tissue level mechanism of this phenomenon, and include quantitative analyses of IHC figures.

R2: Thank you for pointing this out. Following the reviewer’s comment, the following testings and results have been added to the revised manuscript to illustrate the cellular-level mechanism of the beneficial effects of probiotics.

[line 410 to 414]

“ 4.6. Cell viability

2500 BXPC-3 cells were seeded in 96 well-plates at 37°C overnight. Cells were treated with probiotic lysates of GMNL-89 or GMNL-133 in the presence or absence of gemcitabine for 0, 1, 2, 4, 6 days. Cell viability was assessed by 3-(4,5-dimethylthiazol-2yl)-2,5-diphenyltetrazolium bromide (MTT) assay. ”

[line 192 to 198] and Figure 8

2.5. The lysates of probiotics significantly decreased pancreatic cancer cell viability

To further illustrate the effect of GMNL-89 and GMNL-133 on the sensitivity of cancer cells in vitro, the cell viability was measured via MTT assay. The cell viability in the standard chemotherapeutic agent gemcitabine (GCB) combined with probiotic lysate (GMNL-89 + GMNL-133) group was markedly decreased compared to the standard chemotherapy (P<0.001) (Figure 8A). Probiotic lysate (GMNL-89 + GMNL-133) alone also showed the effect of reducing the viability of cancer cells compared to the control group (P<0.001) (Figure 8B).

[line 313 to 324]

In addition to immune regulation, probiotics can also eliminate harmful pathogens through non-immune mechanisms such as lowering the pH of the lumen, competing for nutrients and producing bacteriocin-like substances [39]. The previous report indicated that in patients with atopic dermatitis after taking Lactobacillus paracasei GMNL-133 for three months, the number of Bifidobacterium in stool will be significantly higher than that in the placebo group [40]. Bifidobacterium has been shown to improve therapeutic activity of anti-PD-L1 treatment in the report of animal experiments [41]. In our study, we have also observed the inhibition of probiotic strains on pancreatic cancer cells in vitro. The use of Lactobacillus reuteri GMNL-89 plus Lactobacillus paracasei GMNL-133 with the standard chemotherapeutic agent GCB to inhibit cancer cells is better than the use of single bacteria or only chemotherapy. Even if Lactobacillus reuteri GMNL-89 is used alone without the GCB, it still has the effect of reducing the cancer cells. If it is used in combination with Lactobacillus paracasei GMNL-133, it will have a better inhibitory effect.

Regarding the research on the carcinogenicity of Porphyromonas gingivalis in cell lines, not long ago, we have begun to use BxPC-3 cells for further research. The preliminary results showed that IL-1β, TNF-α, CCL-20 and COX-2 level were increased in P. gingivalis-cultured broth-treated BxPC-3 cells. Other biomarker testing is also underway. However, we still need some time to confirm because the current results are limited. We look forward to more complete results and information to be announced to you in the future.

Point 3. The fact that sequential application of lactobacillus antagonizes P gingivalis virulence indirectly indicate that the overall bacterial load is not a cause for PDAC formation in this model. Some data on plasma or local immune profiles along the study course may help clarify if modulation of host immune activity is a critical component. Composition of tumor infiltrating leucocytes will be especially useful to demonstrate if P gingivalis activates a regulatory immune microenvironment to allow progression of PIN to PDAC.

R3: We have previously performed CD8+ cells detection on pancreatic tissues in three groups: non-treated, P. gingivalis-treated, and P. gingivalis + probiotics-treated. The results show that the P. gingivalis + probiotics-treated group has the least number of CD8+ cells. However, the number of cases in each group is only 1, which is not enough to form a conclusion. Due to the timing of the sacrifice, we could not understand whether the lowest CD8+ T cells in pancreatic tissue of mouse in the P. gingivalis + probiotics-treated group was caused of the lowest severity. 

We had checked the effects of P. gingivalis-cultured broth on cytokine production of immune cells by using of cell lines Jurkat T (mimic T cells) and RAW264.7 (a macrophage-like cell line), our unpublished data indicated that several proinflammatory cytokine expressions were upregulated. As the clinical findings, P. gingivalis triggers periodontitis. We strongly suggest that P. gingivalis and probiotics must contribute to the level of inflammation in pancreas and to the immunoregualtion.

Thank you for pointing out the importance of this part. We will plan the testing and research for this part as soon as possible based on your suggestions.

Point 4. Finally, the authors probably want to discuss what is the virulence mechanism for P gingivalis, which spends most of the life cycle as an intracellular microbe, to promote PDAC formation from PIN. Although experimental proof of such mechanism is beyond the scope of this paper, this will provide an unique angle to hint on a novel perspective of carcinogenesis of this deadly disease and possibly evoke new intervention strategy.

R4: Thank you for your valuable comments. We have previously used the PCR analysis method of Yoneda et al. to confirm the presence of P. gingivalis in the pancreatic tissue of KC mice [1]. However, we have not been able to determine whether it is intracellular. This study illustrates the carcinogenic effects of P. gingivalis in pancreatic cancer and the possibility of probiotic-based treatment options. We are very grateful for the good suggestions provided by the reviewer. We also agree that these experimental results will further strengthen our conclusions. However, these experimental results are too late to be included in the scope of this paper because limited response time and laboratory resources constraints. There are still some questions to be clarified, and so far we can only show preliminary experimental results with a limited sample size. We know that these results are not as comprehensive as the experiments suggested by the reviewer but it still represents our efforts to strengthen the experimental conclusions.

Again, we thank the reviewers and editors for their time and effort to improve our research.

Reference

  1. Yoneda M, Naka S, Nakano K, Wada K, Endo H, Mawatari H, et al. Involvement of a periodontal pathogen, Porphyromonas gingivalis on the pathogenesis of non-alcoholic fatty liver disease. BMC Gastroenterol 2012;12:16.

Sincerely,

Ming-Shiou Jan, PhD 

Attached is the revised article

Reviewer 2 Report

The study showing cancer promoting effect of Porphyromonas gingivalis is interesting and supports the finding from some of the epidemiological studies. However, authors need to provide additional experimental evidence to support their findings.

  1. PanIN lesions are not uniformly distributed in the pancreas. These are also not at the same stage of progression. Therefore, additional parameters which could represent  changes in whole pancreas or significant portion of the tissue are required.
  2. Oral application of Porphyromonas gingivalis was used in the experimental setting, but it is not clear if the observed effect in treated group, is because of established gingivitis or infected pancreatic tissue or changes in gut microbiome or systemic inflammatory changes.
  3. The mechanism of Lactobacillus protective effect in the study is not clear. Changes in gut microbiome and systemic inflammation should be evaluated.
  4. The observed effect, supported by additional robust evaluation and some insight into mechanism will make it valuable.

Author Response

Samuel C. Mok, Ph.D.,

Editor-in-Chief

Cancers

Oct 29, 2020

Dear Prof. Mok,

Thank you very much for the helpful comments on our article (cancers-975891) entitled “Lactobacillus attenuate the progression of pancreatic cancer promoted by Porphyromonas gingivalis in K-rasG12D transgenic mice”. We acknowledge the reviewer’s comments, which are valuable in improving the quality of our manuscript. We have revised our manuscript accordingly and highlighted the changes in blue. In addition, we respond to the editorial board's comments point-by-point as listed below:

Reviewer 2

Point 1.   PanIN lesions are not uniformly distributed in the pancreas. These are also not at the same stage of progression. Therefore, additional parameters which could represent changes in whole pancreas or significant portion of the tissue are required.

R1: You have raised an important point. We have pointed out the limitation of this research design in the discussion part [line 350 to 354] of the manuscript to remind readers. Compared with the complexity and duration of human tumor development, the mouse model is relatively simplified. Whether our research results can be expanded still needs to be tested on different animal models. “Tet-On” system has become an applied tool to control gene activity. We are considering applying this method to experiment to improve this problem. However, with limited experience and funds, we need to spend some time to evaluate the feasibility. Thank you for pointing out the importance of this part.

Point 2.   Oral application of Porphyromonas gingivalis was used in the experimental setting, but it is not clear if the observed effect in treated group, is because of established gingivitis or infected pancreatic tissue or changes in gut microbiome or systemic inflammatory changes.

R2: We have previously used the PCR analysis method of Yoneda et al. to confirm the presence of Porphyromonas gingivalis in the pancreatic tissue of KC mice [1]. We agree with the reviewer that the influence of pancreatic microbiota or gut microflora is very important and needs further research. In addition, we have previously performed CD8+ cells detection on pancreatic tissues in three groups: non-treated, P. gingivalis-treated , and P. gingivalis+probiotics-treated. The results show that the P. gingivalis + probiotics-treated group has the least number of CD8+ cells. However, the number of cases in each group is only 1, which is not enough to form a conclusion. Due to the timing of the sacrifice, we could not understand whether the lowest CD8+ T cells in pancreatic tissue of mouse in the P. gingivalis + probiotics-treated group was caused of the lowest severity.  We will plan further research as soon as possible based on your suggestions.

We totally agree that the reviewer pointed out the issues mentioned above that need to be strengthened. Following the reviewer’s comments, the following text has been added to the revised manuscript to illustrate the association between P. gingivalis and gut microbiota.

[line 288 to 293]

“Previous animal experiments indicate that oral administration of P. gingivalis can cause changes in the intestinal flora [28,29]. Related studies also show that Lactobacillus and Bifidobacterium may have the ability to inhibit P. gingivalis [30,31]. A review study pointed out that there is a correlation between pancreatic cancer and P. gingivalis, and chronic pancreatitis has lower intestinal Lactobacillus and Bifidobacterium [32]. Here we infer that the administration of Lactobacillus may be a strategy to improve pancreatic inflammation and prevent pancreatic cancer.”

Reference

  1. Yoneda M, Naka S, Nakano K, Wada K, Endo H, Mawatari H, et al. Involvement of a periodontal pathogen, Porphyromonas gingivalis on the pathogenesis of non-alcoholic fatty liver disease. BMC Gastroenterol 2012; 12: 16.

Point 3.   The mechanism of Lactobacillus protective effect in the study is not clear. Changes in gut microbiome and systemic inflammation should be evaluated.

R3: Thank you for your valuable suggestions. The description of the immune regulation mechanism of Lactobacillus in the previous literature is mentioned in our discussion [line 293 to 313]. Now, we add a description of the non-immune mechanism of probiotics in the revised draft as listed below:

[line 313 to 324]

In addition to immune regulation, probiotics can also eliminate harmful pathogens through non-immune mechanisms such as lowering the pH of the lumen, competing for nutrients and producing bacteriocin-like substances [39]. The previous report indicated that in patients with atopic dermatitis after taking Lactobacillus paracasei GMNL-133 for three months, the number of Bifidobacterium in stool will be significantly higher than that in the placebo group [40]. Bifidobacterium has been shown to improve therapeutic activity of anti-PD-L1 treatment in the report of animal experiments [41]. In our study, we have also observed the inhibition of probiotic strains on pancreatic cancer cells in vitro. The use of Lactobacillus reuteri GMNL-89 plus Lactobacillus paracasei GMNL-133 with the standard chemotherapeutic agent GCB to inhibit cancer cells is better than the use of single bacteria or only chemotherapy. Even if Lactobacillus reuteri GMNL-89 is used alone without the GCB, it still has the effect of reducing the cancer cells. If it is used in combination with Lactobacillus paracasei GMNL-133, it will have a better inhibitory effect.

At the same time, we add our results regarding the beneficial effect of Lactobacillus on pancreatic cancer cells in vitro.

[line 410 to 414]

“ 4.6. Cell viability

2500 BXPC-3 cells were seeded in 96 well-plates at 37°C overnight. Cells were treated with probiotic lysates of GMNL-89 or GMNL-133 in the presence or absence of gemcitabine for 0, 1, 2, 4, 6 days. Cell viability was assessed by 3-(4,5-dimethylthiazol-2yl)-2,5-diphenyltetrazolium bromide (MTT) assay. ”

[line 192 to 198] and Figure 8

2.5. The lysates of probiotics significantly decreased pancreatic cancer cell viability

To further illustrate the effect of GMNL-89 and GMNL-133 on the sensitivity of cancer cells in vitro, the cell viability was measured via MTT assay. The cell viability in the standard chemotherapeutic agent gemcitabine (GCB) combined with probiotic lysate (GMNL-89 + GMNL-133) group was markedly decreased compared to the standard chemotherapy (P<0.001) (Figure 8A). Probiotic lysate (GMNL-89 + GMNL-133) alone also showed the effect of reducing the viability of cancer cells compared to the control group (P<0.001) (Figure 8B).

[line 313 to 324]

In addition to immune regulation, probiotics can also eliminate harmful pathogens through non-immune mechanisms such as lowering the pH of the lumen, competing for nutrients and producing bacteriocin-like substances [39]. The previous report indicated that in patients with atopic dermatitis after taking Lactobacillus paracasei GMNL-133 for three months, the number of Bifidobacterium in stool will be significantly higher than that in the placebo group [40]. Bifidobacterium has been shown to improve therapeutic activity of anti-PD-L1 treatment in the report of animal experiments [41]. In our study, we have also observed the inhibition of probiotic strains on pancreatic cancer cells in vitro. The use of Lactobacillus reuteri GMNL-89 plus Lactobacillus paracasei GMNL-133 with the standard chemotherapeutic agent GCB to inhibit cancer cells is better than the use of single bacteria or only chemotherapy. Even if Lactobacillus reuteri GMNL-89 is used alone without the GCB, it still has the effect of reducing the cancer cells. If it is used in combination with Lactobacillus paracasei GMNL-133, it will have a better inhibitory effect.

The influence of pancreatic microbiota or gut microflora is very important. We are currently conducting research on intestinal microbial metabolites, including acetic acid, propionic acid, butyric acid, and pentanoic acid, in this mouse model. The preliminary results show that the P. gingivalis-treated group has a high level of butyric acid. In addition, we have also observed that the microbial metabolites of P. gingivalis-treated and P. gingivalis+probiotics-treated groups are very different. The immunomodulatory significance represented by this result and the correlation of the changes in intestinal bacteria still needs more data to clarify.

Point 4.   The observed effect, supported by additional robust evaluation and some insight into mechanism will make it valuable.

R4: We are very grateful for the good comments provided by the reviewer. We also agree that these experimental results will further strengthen our conclusions. However, these experimental results are too late to be included in the scope of this paper because limited response time and laboratory resources constraints. There are still some questions to be clarified, and so far we can only show preliminary experimental results with a limited sample size. We know that these results are not as comprehensive as the experiments suggested by the reviewer but it still represents our efforts to strengthen the experimental conclusions. Our team has also discussed the possibility of future experiments based on your suggestions.

Again, we thank the reviewers and editors for their time and effort to improve our research.

Sincerely,

Ming-Shiou Jan, PhD 

Round 2

Reviewer 1 Report

The addition of cell experiments indicate probiotic lysate may synergize with gemcitabine for inducing early cell death in pancreatic cancer cell line. This provides a mechanistic support for the effects seen in the animal model. Certainly more mechanistic investigation is encouraged following the current work, including quantitative analysis of the potential signaling pathway involved in the pathogenic mechanism. 

Author Response

Point 1. The addition of cell experiments indicate probiotic lysate may synergize with gemcitabine for inducing early cell death in pancreatic cancer cell line. This provides a mechanistic support for the effects seen in the animal model. Certainly more mechanistic investigation is encouraged following the current work, including quantitative analysis of the potential signaling pathway involved in the pathogenic mechanism. 

R1: On the basis of the current research results, we will continue to carry out quantitative analysis of potential signal pathways. Thank you for your valuable suggestions and encouragement.

Reviewer 2 Report

The revised version of the manuscript has expanded discussion and additional published references. The earlier concerns about PanIN lesions and their progression haven't been addressed. Additional parameters of  evaluation of pancreatic histology or analysis should be included to address uneven distribution of PanIN and their progression.

There is no data, about the establishment of Porphyromonas infection or on changes in gut microbiome to understand the link between treatment and resulting adverse progression and its subsequent amelioration by probiotic treatment.

Author Response

Point 1.   The revised version of the manuscript has expanded discussion and additional published references. The earlier concerns about PanIN lesions and their progression haven't been addressed. Additional parameters of evaluation of pancreatic histology or analysis should be included to address uneven distribution of PanIN and their progression.

R1: We present the change in pancreatic weight in Figure 4B and the comparison of PanIN lesions in the newly added Figure 4C, both of which indicate that damage caused by infection is indeed formed after exposure to P. gingivalis and the protective effect of Lactobaccillus. We have added the following texts to the revised manuscript and highlighted the second change in red.

[line 147 to 151]

The quantitative analysis of the number of normal pancreatic ducts and the number of PanIN lesions at different grade are shown in Figure 4C. 12-week old KC mice that were treated with P. gingivalis and a combination of 4-week probiotics feeding had a significantly reduced PanIN 3 lesions compared with the group treated with P. gingivalis without probiotics (P=0.0006).

[line 168 to 174]

C, Number of normal ducts and PanIN lesions were counted and compared between the non-treated, P. gingivalis-treated, and P. gingivalis plus probiotic-treated mice at 18 weeks. The number of normal ducts for the P. gingivalis mice was significantly reduced compared with the non-treated mice (P=0.0002). The number of PanIN 3 lesions for the P. gingivalis plus probiotic-treated mice was significantly reduced compared with the P. gingivalis-treated mice (P=0.0006). Bar graph shows the mean duct number± standard deviation (SD) of a mouse pancreatic tissue section at 100x magnification, n = 10.”

[line 433 to 439]

“4.7. Histologic evaluation of pre-cancer leasions (PanINs)

The formalin-fixed paraffin-embedded pancreatic tissue was sectioned and stained with hematoxylin and eosin, and the histological evaluation of each pancreas section was performed by a pathologist. To quantify the progression of PanIN lesions, we determined the total number of ductal lesions and their grade. According to the naming and classification system of pancreatic duct lesions described in reference 52, each counted cluster is classified into normal ducts, PanIN-1A, PanIN-1B, PanIN-2 and PanIN-3 [52].”

Point 2.   There is no data, about the establishment of Porphyromonas infection or on changes in gut microbiome to understand the link between treatment and resulting adverse progression and its subsequent amelioration by probiotic treatment.

R2:

We cannot add an analysis of the changes in the intestinal flora of our specimen here because the intestinal specimen is not completely preserved. After oral administration of P. gingivalis or Lactobaccillus, the changes in mouse intestinal bacteria and immune regulation have been discussed and verified [ref 28,29,33]. The current study is based on the above conclusions, and mainly focuses on the damage and impact on the pancreas. The key is that we did observe a significant change in pancreatic weight and the grade of PanIN. The conclusion of this article is credible even if there is no evidence of changes in intestinal flora.

To explore the causal relationship between complex bacteria and bacteria or get some conclusions, in addition to the analysis of gut microbiota, a series of metabolites study is inevitable. We still need some time to proceed and raise new experimental mice because our resources are limited. We are sorry for not being able to submit the information requested by the reviewer this time. We agree that the reviewer emphasized the importance of intestinal microbiota research. Therefore, we decide to add a note to the Discussion of revised manuscript to remind this concern about the impact of P. gingivalis and probiotics on gut microbiota.

[line 301 to 302]  

“Previous animal experiments indicate that oral administration of P. gingivalis can cause changes in the intestinal flora and increase the risk of various diseases by changing its metabolic profiles [28,29].”

[line 305to 307]

“Liu et al. reported the effect of Lactobacillus reuteri on the intestinal flora of healthy mice [33]. In addition to the proliferation of beneficial intestinal flora, it also increases the levels of tryptophan metabolites and purine nucleoside adenosine to improve tolerance to inflammatory stimuli.”

[line 376 to 378]

“It is necessary to further analyze the impact of P. gingivalis infection on the intestinal microbiome and host-microbe interactions to understand the relationship between cancer progression and the potential anti-cancer effects of probiotics in humans.”

Round 3

Reviewer 2 Report

The quality of the manuscript has improved with inclusion of additional data. In 'simple summary', authors claim that oral exposure to P. gingivalis can induce carcinogenesis. This statement needs to be modified because presented data suggest that carcinogenesis is accelerated in genetically susceptible mice only. Although authors have included additional literature references to support their observations, in absence of data on establishment of oral infection or its impact on gut microbiome, the conclusions need to be reworded.

In the experiments evaluating the effect of bacterial lysate on cancer cells, please indicate the CFU/ml for preparing lysate and quantity/volume of lysate used per ml of cancer cell media.

Author Response

POINT 1.  The quality of the manuscript has improved with inclusion of additional data. In 'simple summary', authors claim that oral exposure to P. gingivalis can induce carcinogenesis. This statement needs to be modified because presented data suggest that carcinogenesis is accelerated in genetically susceptible mice only.

R1: We have revised the following texts to the revised manuscript and highlighted the change in green.

[line 37 to 39]

We demonstrated that oral exposure to P. gingivalis can accelerate the development of pancreatic ductal adenocarcinoma in mouse model.”

POINT 2.  Although authors have included additional literature references to support their observations, in absence of data on establishment of oral infection or its impact on gut microbiome, the conclusions need to be reworded.

R2: We have reworded the following texts to the revised manuscript.

[line 446 to 450]

We demonstrated that oral exposure to P. gingivalis can accelerate PanIN lesions in KC mice with a severity similar to that caused by caerulein. This clearly shows that P. gingivalis plays a direct role in the deterioration of PDAC. Our findings revealed that Lactobacillus had beneficial effects and could reduce the number and grades of PanIN lesions and the growth and survival of pancreatic cancer cells, inhibit EMT (suppress malignant cell transformation and metastasis) in cancer cells.

[line 452 to 454]

The inhibitory effect of probiotics on PD-L1 expression may be considered as a promising adjuvant treatment to immune checkpoint inhibitors for selected patients with pancreatic cancer in the future.

POINT 3. In the experiments evaluating the effect of bacterial lysate on cancer cells, please indicate the CFU/ml for preparing lysate and quantity/volume of lysate used per ml of cancer cell media.

R3: We have added the relevant instructions to the revised manuscript.

[line 434 to 436]

Cells were treated with probiotic lysates of GMNL-89 (25µg/mL; 6.41 x 108 CFU/mL) or GMNL-133 (25µg/mL; 7.76 x 108 CFU/mL) in the presence or absence of gemcitabine for 0, 1, 2, 4, 6 days.”
